# PERTURBATION DISCRIMINATION-ENHANCED GRAPH CONTRASTIVE LEARNING

## ABSTRACT

Self-supervised learning of graph-structured data aims to produce transferable and robust representations that could be transferred to the downstream tasks. Among many, graph contrastive learning (GCL) based on data augmentation has emerged with promising performance in learning graph representation. However, it is observed that some augmentations might change the graph semantics due to the perturbations in the graph structure such as perturbing some nodes/edges. In such cases, existing GCL methods may suffer from performance limitations due to the introduction of noise augmentations. To address this issue, we propose to train a discriminative model to enhance GCL for graph-structured data, called Perturbation Discrimination-Enhanced GCL (PerEG). Specifically, for each perturbed graph, the discriminative model is trained to predict whether each node in the augmentation was perturbed by the perturbation compared to the original graph or not. Based on this, the results of perturbation discrimination are exploited to refine the GCL, enabling its controllable use of augmentation, thereby preferably utilizing augmentation and effectively avoiding the introduction of noise augmentation. Extensive experiments in unsupervised, semi-supervised, and transfer learning scenarios show that our PerEG outperforms the state-of-the-art methods on eight datasets.

## 1 INTRODUCTION

Graph neural networks (GNNs) (Kipf & Welling, 2017; Velickovic et al., 2018; Xu et al., 2019) are becoming increasingly popular in the realm of learning graph representation for graph-structured data, where the structural information of graphs could be modeled with a neighborhood aggregation scheme. By inheriting the power of neural networks, GNNs have reached great progress in learning information from real-world graphs such as knowledge graphs (Baek et al., 2020), social networks (Fan et al., 2019), point clouds (Shi & Rajkumar, 2020), and chemical analysis (De Cao & Kipf, 2018). Most existing GNN-based models are trained in a supervised manner with end-to-end form when solving graph-based tasks. However, annotating a high volume of fine-annotated data requires time-consuming laboratory annotations based on professional knowledge (Hu et al., 2020a; Tan et al., 2021).

To alleviate this issue, various self-supervised learning methods are explored on graph-structured data for learning transferable and robust representations from unlabeled graphs (Hu et al., 2020a; You et al., 2020). Among them, graph predictive learning and graph contrastive learning (GCL) have received a lot of attention in learning graph representations. Here graph predictive learning can learn the contextual relationships between neighboring nodes (Hu et al., 2020a; Kim & Oh, 2021) or discriminating original graphs from perturbed graphs (Kim et al., 2022), and graph contrastive learning (GCL) (You et al., 2020; Xu et al., 2021; Xia et al., 2022b) can learn the mutual information between augmentations by maximizing the similarity between augmented views. While promising progress is made, existing methods generally attempt to learn the differences or similarities from the graph augmentations obtained by augmentation strategy link perturbations on nodes or edges. However, as many studies have pointed out, the perturbation of nodes/edges of the graph might change the nature of the original graphs (Kim et al., 2022; Yue et al., 2022). In addition, another study attempts to perturb the encoder to obtain contrastive objectives without any change in the graph structure, to preserve the graph semantics well (Xia et al., 2022a). However, this work is more like a general method at the representation level, ignoring the learning of rich structured information when performing GCL. We thus argue that to explore rich structural information for improving graph representation, we should use graph augmentation with structural perturbations, but it is necessary to

control the application of augmentation to graph contrastive learning by identifying the properties of perturbations to prevent the introduction of noise.

To reach this goal, we propose a discriminative model to identify the fine-grained perturbation in a perturbed augmentation compared to the original graph, and then enhance GCL based on the discriminative results, called Perturbation Discrimination-Enhanced Graph Contrastive Learning (PerEG). We observe that for an augmentation of an original graph, its structural perturbations should be effective and tolerable, that is, the augmentation is different from but associated with the original graph. Therefore, for a qualified augmented graph, the model should be able to identify where it has been disturbed compared to the original graph. Otherwise, if a perturbed graph is completely different from the original graph, it probably changes the semantic information of the original graph and cannot be used as an augmentation of the original graph for learning graph representation.

In light of this, we get inspiration from ELECTRA (Clark et al., 2019) and train a discriminative model that predicts whether each node in the augmented graph was affected by perturbations or not, aiming to perceive the fine-grained perturbation of a perturbed augmentation compared to the original graph. The motivation for this operation comes from the fact that a vertex/node is the fundamental unit of which graphs are formed, and judging changes in node representation can identify where the augmentation has been disturbed compared to the original graph. Further, the perturbation of a node is usually caused by two situations: one is directly perturbing the node, and the other is caused by the perturbation of the edge. To this end, we devise our discriminative model from two perspectives: node-oriented discriminator and edge-oriented discriminator. In this way, both node representation changes caused by node perturbation and by edge perturbation will be taken into account. It should be noted that if perturbed nodes in an augmentation cannot be accurately identified, their representations are either very similar to the original ones or have deviated significantly in the latent space. In such cases, those augmentations cannot be considered as qualified perturbed augmentations and should be limited in GCL, because they are neither effective nor plausible. Subsequently, the discriminative accuracy of perturbed nodes is exploited in the GCL, which essentially enables the model to use graph augmentation in a controlled manner, effectively avoiding the introduction of noise augmentation. Our main contributions are as follows:

- We propose Perturbation Discrimination-Enhanced Graph Contrastive Learning (PerEG), the first to enhance GCL by training a discriminative model for perturbation identification, which enables the model to use graph augmentations for GCL in a controlled manner, effectively avoiding the introduction of noise augmentation.

- Instead of laboriously entangling about how to prevent the introduction of noise structures during graph augmentation, this paper aims to find the potential noise augmentations that might change the graph semantics and limit them in GCL according to the results of perturbation discrimination.

- Node-oriented and edge-oriented discriminators are designed to identify whether each node in the perturbed graph is affected by the perturbations or not, empowering our PerEG to discriminate the effective and plausible augmented graphs for contrastive learning.

- We conduct a series of experiments on eight graph-structured benchmark datasets to evaluate the effectiveness of our PerEG in both unsupervised and semi-supervised scenarios. Experimental results show that our PerEG significantly outperforms baselines.

## 2 RELATED WORK

### 2.1 GRAPH NEURAL NETWORKS

Graphs are a kind of prevalent data structure in real-world scenarios, such as molecule graphs, social networks, and biological graphs (Zhou et al., 2020). Modeling the set of objects (nodes) and their relationships (edges) of graph-structured data is an appropriate way to learn the representation of the graph structure (Hamilton et al., 2017; Xu et al., 2019). GNNs are a practical framework for graph representation learning. Owing to the neighborhood aggregation scheme of GNNs, the representation vectors of nodes are calculated by recursively aggregating and transforming the representation vectors of their adjacent nodes (Gilmer et al., 2017). Recently, there has been a surge of interest in obtaining better graph representations, and various GNN architectures have been proposed for updating and

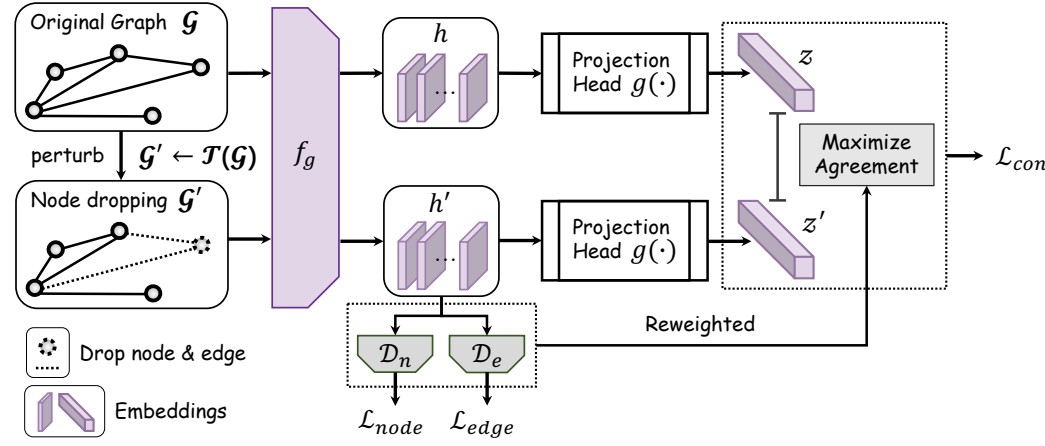

Figure 1: Illustration of our PerEG framework. The structure-perturbed graph $\mathcal{G}'$ is an augmentation of $\mathcal{G}$ sampled from an augmentation pool $\mathcal{T}(\mathcal{G})$. $f_g$ is a GNN encoder. $\mathcal{D}_n$ is the node-oriented discriminator and $\mathcal{D}_e$ is the edge-oriented discriminator. $\mathcal{L}_{con}$ represents the contrastive loss. $\mathcal{L}_{node}$ and $\mathcal{L}_{edge}$ represent the discriminative losses of $\mathcal{D}_n$ and $\mathcal{D}_e$, respectively.

aggregation of graphs. To mention a few, Graph Convolutional Network (GCN) (Kipf & Welling, 2017) utilizes an approximation of spectral graph convolutions to learn representations that encode both local graph structure and features of nodes.

## 2.2 SELF-SUPERVISED LEARNING FOR GRAPH REPRESENTATION

Many studies have been devoted to graph representation learning based on self-supervised learning (Hu et al., 2020a). Recently, some studies have attempted to perform prediction in learning contextual properties of graph-structured data (Kim & Oh, 2021; Hwang et al., 2020; Hu et al., 2020b; Rong et al., 2020). Further, D-SLA (Kim et al., 2022) proposes to discriminate the original graphs from the perturbed graphs. In addition, many recent studies are dedicated to exploring contrastive learning to achieve better graph representations based on graph augmentations. GraphCL (You et al., 2020) designs four types of graph augmentations to learn the similarly metrics between augmentations with contrastive learning. JOAO (You et al., 2021) and AutoGCL (Yin et al., 2022) propose an auto augmentation strategy to enhance the performance of graph contrastive learning. MVGRL (Hassani & Khasahmadi, 2020) performs contrastive learning on different structural views of graphs for both node and graph levels to enrich the feature learning. In addition, some recent studies attempt to alleviate the issue that some augmentations might change the graph semantics in graph contrastive learning. SimGRACE (Xia et al., 2022a) generates contrastive objects of graphs for contrast by perturbing the weights of the model rather than structure perturbations. However, this work overlooks the exploration of graph structures and thus cannot produce robust graph representations by introducing richer graph structures. Further, GLA (Yue et al., 2022) utilizes label information to keep the label of the augmented sample the same as the original graph for contrast. However, this work requires the introduction of annotations and cannot be used in unsupervised scenarios.

## 3 METHOD

This section introduces our novel PerEG framework in detail. The architecture of PerEG is illustrated in Figure 1. We first recount the preliminaries in section 3.1. Then, we introduce the discriminative model and perturbation discrimination-enhanced graph contrastive learning of our PerEG in Section 3.2 and Section 3.3, respectively. We finally describe the learning objective in Section 3.4.

### 3.1 PRELIMINARIES

**Graph Neural Networks**   Let $\mathcal{G} = \{\mathcal{V}, \mathcal{E}\}$ denote an undirected graph, where $\mathcal{V}$ and $\mathcal{E}$ are the sets of nodes and edges, respectively, $\boldsymbol{X}_v \in \mathbb{R}^{|\mathcal{V}| \times d_v}$ denote the matrix of node features, and $\boldsymbol{X}_e \in \mathbb{R}^{|\mathcal{E}| \times d_e}$ denote the matrix of edge features. Here, the representation of a node $v$ can be defined as $h_v$, and the representation of a graph $\mathcal{G}$ can be defined as $h_{\mathcal{G}}$. The goal of graph neural networks (GNNs) is to update the representation of the given graph $\mathcal{G}$ by leveraging its topological structure. For the representation $h_v$ of node $v$, its propagation of the $k$-th layer GNN is represented as:

$$h_v^{(k)} = f_g(h_v^{(k-1)}; \theta) = \text{UPDATE}^{(k)}\big(h_v^{(k-1)}, \text{AGGREGATION}^{(k)}\big(h_u^{(k-1)} : \forall u \in \mathcal{N}(v)\big)\big) \quad (1)$$

where $f_g(\cdot; \theta)$ denotes the GNN encoder, $\theta$ represents all trainable parameters of GNN encoder. $\text{AGGREGATION}(\cdot)$ denotes a trainable function that aggregates messages from the neighbors of node $v$, $\mathcal{N}(v)$ represents the set of neighbors. $\text{UPDATE}(\cdot)$ denotes a trainable function that updates the representation of node $v$ with the current representation of $v$ and the aggregated vector. Then, after $K$ iterations by GNNs, the representation of a graph $\mathcal{G}$ can be obtained by pooling the final set of all node representations from Eq. 1, which is represented as:

$$h_{\mathcal{G}} = \text{POOL}\big(\{h_v^{(K)} : \forall v \in \mathcal{V}\}\big) \quad (2)$$

The operation of $\text{POOL}(\cdot)$ is flexible, including mean or sum, and other relatively well-designed methods, such as clustering (Ying et al., 2018; Baek et al., 2021) or node drop-based methods (Gao & Ji, 2019; Lee et al., 2019).

**Data Augmentation for Graphs**   Following (You et al., 2020), to explore the rich structure of a graph, we use **Node dropping**, **Edge perturbation**, **Attribute masking**, and **Subgraph** to generate augmentations for each graph. Note that, the purpose of our work is to perform node-level perturbation discrimination. Therefore, we applied four types of perturbations to each graph to alleviate the impact on the effectiveness of perturbation discrimination that a node only appearing in the perturbed set.

**Overall Framework**   As illustrated in Figure 1, the proposed PerEG framework consists of three modules: 1) Node-oriented Discriminator, which predicts whether each node in the augmented graph was perturbed or not by the perturbation acting on nodes; 2) Edge-oriented Discriminator, which predicts whether each node in the augmented graph was affected by edges' perturbations; 3) Reweighted Contrastive Learning, which performs graph contrastive learning reweighted by the perturbation discrimination results.

### 3.2 THE DISCRIMINATIVE MODEL IN OUR PERG

Figure 2 shows the architecture of the discriminative model in our PerEG, which includes a Node-oriented Discriminator $\mathcal{D}_n$ (left) and an Edge-oriented Discriminator $\mathcal{D}_e$ (right). Specifically, the Node-oriented Discriminator aims to identify whether each node was perturbed (dropped or masked) in the augmented graph or not. The Edge-oriented Discriminator aims to identify whether each node is affected by the edges' perturbations (adding or dropping) in the augmentation or not.

As mentioned above, the properties and semantics of graphs might change even with slight perturbations of nodes/edges (Kim et al., 2022; Yue et al., 2022). Therefore, different from using contrastive learning that blindly trains perturbed augmentations of a graph to be similar, we explore a discriminative model to perceive the perturbations of the augmented view compared to the original graph, aiming to control contrastive learning through perturbation discrimination results. The motivation comes from

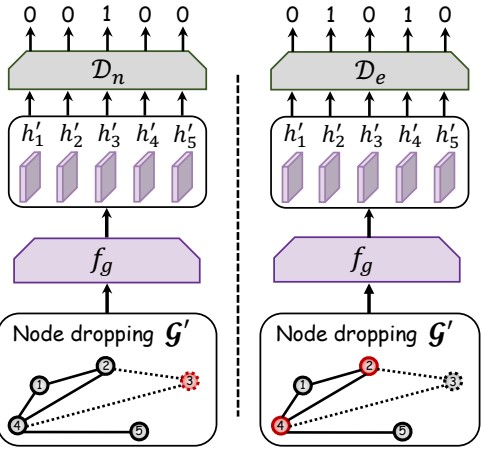

Figure 2: Illustration of the discriminative model in our PerEG framework.

the fact that suppose a perturbed graph can be used as an augmentation for contrastive learning. In that case, it must be related to the original graph, that is, the perturbations should be identifiable.

To reach this goal, we propose to discriminate the perturbations based on predicting whether each node in the augmented graph was affected by perturbations or not. Because node representations are the sources for producing graph representation, predicting the perturbations at the node level can lead to a clear understanding of the graph's transformations. Intuitively, the perturbation of a node can come from the perturbation of the node itself (direct perturbation) or from the impact caused by edge perturbations (indirect perturbation). Therefore, we propose to train the discriminate model from node-oriented and edge-oriented perspectives, devising node-oriented and edge-oriented discriminators.

**Node-oriented Discriminator**  The goal of the Node-oriented Discriminator $\mathcal{D}_n$ is to predict whether each node is perturbed (dropped or masked) in the augmented graph or not, so as to perceive which perturbations the original graph has undergone to generate the augmented view. Here, we feed each node representation $h_v^{'}$ learned from the augmentation into $\mathcal{D}_n$ and employ a cross-entropy loss to train $\mathcal{D}_n$. The loss of $\mathcal{D}_n$ is defined as:

$$\mathcal{L}_{node} = \sum_{v \in \mathcal{V}} \Big( -\mathbb{1}\big(\mathcal{P}_n(v)=1\big)\log\big(\mathcal{D}_n(h_v^{'};\theta_n)\big) - \mathbb{1}\big(\mathcal{P}_n(v)=0\big)\log\big(1-\mathcal{D}_n(h_v^{'};\theta_n)\big) \Big) \quad (3)$$

where $\mathbb{1}(i=j) \in \{0,1\}$ is an indicator function evaluating to 1 iff $i=j$. $\mathcal{P}_n(v)=1$ denotes node $v$ is perturbed in the graph, and $\mathcal{P}_n(v)=0$ is the opposite. $\theta_n$ represents the trainable parameters of the Node-oriented Discriminator $\mathcal{D}_n$.

**Edge-oriented Discriminator**  Predicting the perturbed nodes can directly discriminate the fine-grained perturbation of the graph, however, the structure-perturbed augmentation might create isolated outlier nodes that share similar topological formulas even across different graphs, we thus propose to predict the neighbors of perturbed nodes/edges from the edge-oriented perspective to identify the topological perturbation of the augmentation. As illustrated in Figure 2, in addition to the perturbing node (node 3), the neighbors (nodes 2 and 4) are potentially affected owing to the edges connected to node 3. Therefore, we propose an Edge-oriented Discriminator $\mathcal{D}_e$ to identify whether each node is affected by the edges' perturbations or not. The loss of $\mathcal{D}_e$ is defined as:

$$\mathcal{L}_{edge} = \sum_{v \in \mathcal{V}} \Big( -\mathbb{1}\big(\mathcal{P}_e(v)=1\big)\log\big(\mathcal{D}_e(h_v^{'};\theta_e)\big) - \mathbb{1}\big(\mathcal{P}_e(v)=0\big)\log\big(1-\mathcal{D}_e(h_v^{'};\theta_e)\big) \Big) \quad (4)$$

where $\mathcal{P}_e(v)=1$ denotes node $v$ is affected by the edges' perturbations, and $\mathcal{P}_e(v)=0$ is the opposite. $\theta_e$ represents the trainable parameters of the Edge-oriented Discriminator $\mathcal{D}_e$.

### 3.3 Reweighted Contrastive Learning in Our PerEG

To relieve the performance degradation caused by noise augmentation in contrastive learning, we then exploit the discriminative results to reweight the GCL, so as to perform the GCL in a controllable manner. The reweighted factor $\rho$ is computed as:

$$\rho_n = \frac{1}{|\mathcal{V}_{pn}|} \sum_{v \in \mathcal{V}_{pn}} \Big( \mathbb{1}(\mathcal{D}_n(v)=1) \Big), \quad \rho_e = \frac{1}{|\mathcal{V}_{pe}|} \sum_{v \in \mathcal{V}_{pe}} \Big( \mathbb{1}(\mathcal{D}_e(v)=1) \Big), \quad \rho = \gamma_n \rho_n + \gamma_e \rho_e \quad (5)$$

where $\mathcal{V}_{pn}$ represents the set of nodes directly perturbed by perturbation, $\mathcal{V}_{pe}$ represents the set of nodes affected by the edges' perturbations. $\gamma_n$ and $\gamma_e$ are the coefficients of reweighted factor $\rho$. Here, we only consider the discriminative success rate of the disturbed nodes, due to the goal of the proposed PerEG is to enhance the GCL based on the perturbation discrimination. Further, for those perturbed nodes that have not been identified, they either learn to be similar to the original node or have completely deviated from the graph structure. In such cases, these nodes are detrimental to augmentation. The purpose of augmentation is to explore richer graph structures by performing some reasonable perturbations on the original graph. Therefore, these augmentations should be different from the original graph, but not completely unrecognizable.

For contrastive training, we first randomly select $N_b$ graphs $\{\mathcal{G}_i\}_{i=1}^{N_b}$ from a given dataset to obtain a minibatch $\mathcal{B}$. For an *anchor* graph $\mathcal{G}_i$ and the corresponding perturbed augmentation $\mathcal{G}_i^{'}$, two

representations $z_i$ and $z_i^{'}$ can be obtained from the GNN encoder $f_g(\cdot; \theta)$, which are considered as a *positive* pair. While the representations of the other $N_b - 1$ graphs $\{\mathcal{G}_k \in \mathcal{B}, k \neq i\}$ in the same minibatch $\mathcal{B}$ are treated as *negative* representations with respect to the *anchor* graph $\mathcal{G}_i$ as in (Chen et al., 2017; 2020; You et al., 2020; Xia et al., 2022a). Then, based on the reweighted factor $\rho$, the contrastive loss is computed across all *positive* pairs in a mini-batch $\mathcal{B}$, which is defined as:

$$\mathcal{L}_{con} = \frac{1}{N_b} \sum_{\mathcal{G}_i \in \mathcal{B}} \ell(\mathcal{G}_i), \quad \ell(\mathcal{G}_i) = -\log \frac{\exp(\text{sim}(z_i, z_i^{'})/\tau)}{\sum_{k=1, k \neq i}^{N_b} \exp(\text{sim}(z_i, z_k)/\tau)} \times \rho \tag{6}$$

where $\ell(\mathcal{G}_i)$ denotes the NT-Xent for graph $\mathcal{G}_i$. $\text{sim}(z_i, z_k) = z_i^{\top} z_k / \|z_i\| \|z_k\|$ denotes the cosine similarity function. $\tau$ denotes the temperature parameter.

### 3.4 Overall Learning Objective of Our PerEG

The learning objective of our PerEG is to train the framework by jointly minimizing the three losses derived from Node-oriented Discriminator, Edge-oriented Discriminator, and Reweighted Contrastive Learning. The overall loss $\mathcal{L}$ is defined as:

$$\mathcal{L} = \lambda_1 \mathcal{L}_{con} + \lambda_2 \mathcal{L}_{node} + \lambda_3 \mathcal{L}_{edge} \tag{7}$$

where hyperparameters $\lambda_1$, $\lambda_2$ and $\lambda_3$ are scaling weights to balance the losses.

## 4 Experiments

In this section, we are devoted to the empirical evaluation of the proposed Simple Self-supervised Learning for Graph Representation (PerEG). We first describe the experimental settings of our empirical evaluation. Then, we experimentally validate our PerEG on graph classification tasks to show its effectiveness in producing semantically good graph representations, thus improving the performance of downstream tasks. Finally, we provide various insightful experiments and extensive analysis to demonstrate why our PerEG is valid for producing accurate graph representation.

### 4.1 Experimental Settings

**Datasets** To evaluate the effectiveness of our PerEG framework, we conduct experiments on eight publicly available benchmark datasets from TUDatasets (Morris et al., 2020), including four graph datasets related to biochemical molecules and proteins (MUTAG (Debnath et al., 1991), PROTEINS (Borgwardt et al., 2005), DD (Dobson & Doig, 2003), and

Table 1: Statistics of datasets.

| Datasets | Category | #Graph | Avg. #Node | Avg. #Edge |
|---|---|---|---|---|
| NCI1 | Biochemical Molecules | 4110 | 29.87 | 32.30 |
| PROTEINS | Biochemical Molecules | 1113 | 39.06 | 72.82 |
| DD | Biochemical Molecules | 1178 | 284.32 | 715.66 |
| MUTAG | Biochemical Molecules | 188 | 17.93 | 19.79 |
| COLLAB | Social Networks | 5000 | 74.49 | 2457.78 |
| RDT-B | Social Networks | 2000 | 429.63 | 497.75 |
| RDT-M5K | Social Networks | 5000 | 508.52 | 594.87 |
| IMDB-B | Social Networks | 1000 | 19.77 | 96.53 |

NCI1 (Wale et al., 2008)) and four graph datasets related to social networks (Yanardag & Vishwanathan, 2015) (COLLAB, RDT-B, RDT-M5K, and IMDB-B). The statistics of these datasets are shown in Table 1. Noting that, the datasets are from two categories, and the numbers of graphs in these datasets range from 188 to 5000, the average node numbers range from 17.93 to 508.52, and the average edge numbers range from 19.79 to 2457.78, denoting the diversity of these datasets.

**Evaluation Protocols** Following the same protocols used in existing works for graph self-supervised representation learning in the unsupervised scenario (You et al., 2020; 2021; Xia et al., 2022a), we use the whole dataset for pre-training to learn graph embeddings with our PerEG and feed them into a downstream SVM classifier with 10-fold cross-validation. For the semi-supervised scenario, following (You et al., 2020; 2021; Xia et al., 2022a), we perform pre-training on all the data and later do fine-tuning and evaluation with 10% true label supervision on the same dataset. For fine-tuning, the encoder has an additional linear graph prediction layer on top which is used to map the representations to the task labels. For the transfer learning scenario, we pre-train the encoder on all the data of one dataset and fine-tune and evaluate with another dataset which distinct from the former. The classification accuracy is reported to evaluate the performance.

Table 2: Unsupervised representation learning on TUDataset. Averaged accuracy±std. (%) over 10 runs are reported. "w/o $\mathcal{L}_{node}$" denotes without using edge-oriented discriminator in our PerEG. "w/o $\mathcal{L}_{edge}$" denotes without using edge-oriented discriminator, "w/o $\rho$ denotes without using factor $\rho$. "-" indicates that the label rate is too low for a given dataset size or the baseline did not report the corresponding results. The best and second-best results are highlighted in **red** and **blue**, respectively. A.R. is short for the average rank.

| Methods | NCI1 | PROTEINS | DD | MUTAG | COLLAB | RDT-B | RDT-M5K | IMDB-B | A.R.↓ |
|---|---|---|---|---|---|---|---|---|---|
| GL | - | - | - | 81.66±2.11 | - | 77.34±0.18 | 41.01±0.17 | 65.87±0.98 | 16.5 |
| WL | 80.01±0.50 | 72.92±0.56 | - | 80.72±3.00 | - | 68.82±0.41 | 46.06±0.21 | 72.30±3.44 | 12.8 |
| DGK | 80.31±0.46 | 73.30±0.82 | - | 87.44±2.72 | - | 78.04±0.39 | 41.27±0.18 | 66.96±0.56 | 12.7 |
| node2vec | 54.89±1.61 | 57.49±3.57 | - | 72.63±10.20 | - | - | - | - | 17.7 |
| sub2vec | 52.84±1.47 | 53.03±5.55 | - | 61.05±15.80 | - | 71.48±0.41 | 36.68±0.42 | 55.26±1.54 | 18.5 |
| graph2vec | 73.22±1.81 | 73.30±2.05 | - | 83.15±9.25 | - | 75.78±1.03 | 47.86±0.26 | 71.10±0.54 | 14.2 |
| InfoGraph | 76.20±1.06 | 74.44±0.31 | 72.85±1.78 | 89.01±1.13 | 70.65±1.13 | 82.50±1.42 | 53.46±1.03 | **73.03**±0.87 | 9.9 |
| MVGRL | - | - | - | 75.40±7.80 | - | 82.00±1.10 | - | 63.60±4.20 | 16.7 |
| GraphCL | 77.87±0.41 | 74.39±0.45 | 78.62±0.40 | 86.80±1.34 | 71.36±1.15 | **89.53**±0.84 | 55.99±0.28 | 71.14±0.44 | 8.0 |
| JOAO | 78.07±0.47 | 74.55±0.41 | 77.32±0.54 | 87.35±1.02 | 69.50±0.36 | 85.29±1.35 | 55.74±0.63 | 70.21±3.08 | 10.6 |
| JOAOv2 | 78.36±0.53 | 74.07±1.10 | 77.40±1.15 | 87.67±0.79 | 69.33±0.34 | 86.42±1.45 | 56.03±0.27 | 70.83±0.25 | 9.6 |
| AD-GCL-FIX | 69.67±0.51 | 73.59±0.65 | 74.49±0.52 | 89.25±1.45 | 73.32±0.61 | 85.52±0.79 | 53.00±0.82 | 71.57±1.01 | 9.9 |
| AD-GCL-OPT | 69.67±0.51 | 73.81±0.46 | 75.10±0.39 | 89.70±1.03 | 73.32±0.61 | 85.52±0.79 | 54.93±0.43 | 72.33±0.56 | 8.6 |
| SimGRACE | 79.12±0.44 | 75.35±0.09 | 77.44±1.11 | 89.01±1.31 | 71.72±0.82 | 89.51±0.89 | 55.91±0.34 | 71.30±0.77 | 6.8 |
| AutoGCL | **82.00**±0.29 | 75.80±0.36 | 77.57±0.60 | 88.64±1.08 | 70.12±0.68 | 88.58±1.49 | **56.75**±0.18 | **73.30**±0.40 | 5.4 |
| GPA | 80.42±0.41 | **75.94**±0.25 | **79.90**±0.35 | **89.68**±0.80 | **76.17**±0.10 | 89.32±0.38 | 53.70±0.19 | 74.64±0.35 | **3.5** |
| PerEG (ours) | 79.94±0.24 | **76.04**±0.22 | **80.02**±0.34 | **90.61**±1.50 | **73.84**±0.63 | **90.57**±0.36 | **56.81**±0.24 | 72.97±0.43 | **2.0** |
| w/o $\mathcal{L}_{node}$ | 74.56±0.72 | 75.04±0.50 | 78.60±1.30 | 89.57±0.72 | 71.76±0.71 | 88.86±0.60 | 55.80±0.25 | 71.07±0.32 | 7.8 |
| w/o $\mathcal{L}_{edge}$ | 75.03±0.43 | 75.43±0.39 | 75.67±1.96 | 89.02±0.61 | 71.93±0.13 | 89.23±1.15 | 56.57±0.24 | 71.33±0.40 | 7.0 |
| w/o $\rho$ | 79.69±0.03 | 75.78±0.06 | 78.76±0.57 | 90.13±0.08 | 70.68±0.02 | 90.35±0.01 | 55.50±0.93 | 69.59±0.03 | 6.3 |

Table 3: Semi-supervised learning performance with 10% labels on TUDataset. Averaged accuracy±std. (%) over 10 runs are reported. The best results are highlighted in **red**. A.R. is short for the average rank.

| Methods | NCI1 | PROTEINS | DD | COLLAB | RDT-B | RDT-M5K | A.R.↓ |
|---|---|---|---|---|---|---|---|
| No Pre-Train | 73.72±0.24 | 70.40±1.54 | 73.56±0.41 | 73.71± 0.27 | 86.63±0.27 | 51.33±0.44 | 9.0 |
| GAE | 74.36±0.24 | 70.51±0.17 | 74.54±0.68 | 75.09±0.19 | 87.69±0.40 | 53.58±0.13 | 7.0 |
| InfoGraph | 74.86±0.26 | 72.27±0.40 | 75.78±0.34 | 73.76±0.29 | 88.66±0.95 | 53.61±0.31 | 5.3 |
| ContextPred | 73.00±0.30 | 70.23±0.63 | 74.66±0.51 | 73.69±0.37 | 84.76±0.52 | 51.23±0.84 | 9.5 |
| GraphCL | 74.63±0.25 | 74.17±0.34 | 76.17±1.37 | 74.23±0.21 | 89.11±0.19 | 52.55±0.45 | 5.0 |
| JOAO | 74.48±0.27 | 72.13±0.92 | 75.69±0.67 | 75.30±0.32 | 88.14±0.25 | 52.83±0.54 | 6.3 |
| JOAOv2 | 74.86±0.39 | 73.31±0.48 | 75.81±0.73 | 75.53±0.18 | 88.79±0.65 | 52.71±0.28 | 4.8 |
| AD-GCL-FIX | 75.18±0.31 | 73.96±0.47 | **77.91**±0.73 | **75.82**±0.26 | **90.10**±0.15 | 53.49±0.28 | 2.7 |
| GPA | **75.50**±0.14 | **74.27**±1.11 | 76.68±0.81 | - | 89.99±0.32 | **54.92**±0.35 | **2.0** |
| PerEG (ours) | **75.78**±0.52 | **73.82**±0.58 | 77.23±0.51 | **75.85**±0.51 | **90.12**±0.80 | **54.31**±0.17 | **1.8** |

**Compared Baselines**  We compare our PerEG with various baseline models, including graph kernel methods: GL (Shervashidze et al., 2009), WL (Shervashidze et al., 2011), and DGK (Yanardag & Vishwanathan, 2015), graph self-supervised learning methods: GAE (Kipf & Welling, 2016), node2vec (Grover & Leskovec, 2016), sub2vec (Adhikari et al., 2018), graph2vec (Narayanan et al., 2017), and InfoGraph (Sun et al., 2020), predictive learning method: ContextPred (Hu et al., 2020a), and contrastive learning methods: MVGRL (Hassani & Khasahmadi, 2020), GraphCL (You et al., 2020), JOAO and JOAOv2 (You et al., 2021), AD-GCL-FIX and AD-GCL-OPT (Suresh et al., 2021), SimGRACE (Xia et al., 2022a), AutoGCL (Yin et al., 2022) and GPA (Zhang et al., 2024).

**Implementation Details**  Following previous graph self-supervised learning methods (You et al., 2020), we use the GIN (Xu et al., 2019) as the base network in our PerEG. The augmentation (dropping, perturbation, masking, and subgraph) ratio is set at 0.2 as implemented in GraphCL (You et al., 2020). The discriminator $\mathcal{D}_n$ and $\mathcal{D}_e$ are implemented by a 3-layers MLP. The scaling weights of losses are set to $\lambda_1 = 1$, $\lambda_2 = 1$, and $\lambda_3 = 0.5$, which are the optimal hyper-parameters in the pilot studies.

### 4.2 PERFORMANCE IN GRAPH CLASSIFICATION

**Unsupervised Learning**  Table 2 shows the results of graph classification under the unsupervised scenario. It can be seen that our PerEG outperforms other baselines, achieving the best performance

Table 4: Transfer learning comparison with different manually designed pre-training schemes. ROC-AUC±std. (%) over 10 runs are reported. The best and second-best results are highlighted in **red** and **blue**, respectively. A.R. denotes average rank.

| Pre-Train | ZINC-2M | | | | | | | | PPI-306K | |
|---|---|---|---|---|---|---|---|---|---|---|
| Fine-Tune | BBBP | Tox21 | ToxCast | SIDER | ClinTox | MUV | HIV | BACE | PPI | A.R.↓ |
| No Pre-Train | 65.8±4.5 | 74.0±0.8 | 63.4±0.6 | 57.3±1.6 | 58.0±4.4 | 71.8±2.5 | 75.3±1.9 | 70.1±5.4 | 64.8±1.0 | 8.4 |
| Infomax | 68.8±0.8 | 75.3±0.5 | 62.7±0.4 | 58.4±0.8 | 69.9±3.0 | 75.3±2.5 | 76.0±0.7 | 75.9±1.6 | 64.1±1.5 | 7.0 |
| EdgePred | 67.3±2.4 | 76.0±0.6 | 64.1±0.6 | 60.4±0.7 | 64.1±3.7 | 74.1±2.1 | 76.3±1.0 | **79.9**±0.9 | 65.7±1.3 | 5.3 |
| AttrMasking | 64.3±2.8 | **76.7**±0.4 | **64.2**±0.5 | **61.0**±0.7 | 71.8±4.1 | 74.7±1.4 | 77.2±1.1 | 79.3±1.6 | 65.2±1.6 | **4.3** |
| ContextPred | 68.0±2.0 | 75.7±0.7 | 63.9±0.6 | 60.9±0.6 | 65.9±3.8 | 75.8±1.7 | 77.3±1.0 | **79.6**±1.2 | 64.4±1.3 | 4.8 |
| GraphCL | 69.68±0.67 | 73.87±0.66 | 62.40±0.57 | 60.53±0.88 | 75.99±2.65 | 69.80±2.66 | **78.47**±1.22 | 75.38±1.44 | 67.88±0.85 | 6.2 |
| JOAO | 70.22±0.98 | 74.98±0.29 | 62.94±0.48 | 59.97±0.79 | **81.32**±2.49 | 71.66±1.43 | 76.73±1.23 | 77.34±0.48 | 64.43±1.38 | 6.0 |
| JOAOv2 | **71.39**±0.92 | 74.27±0.62 | 63.16±0.45 | 60.49±0.74 | 80.97±1.64 | 73.67±1.00 | 77.51±1.17 | 75.49±1.27 | 63.94±1.59 | 5.8 |
| SimGRACE | 71.25±0.86 | 75.21±0.93 | 63.36±0.52 | 60.59±0.96 | 75.83±2.63 | **76.86**±1.27 | 75.21±0.87 | 74.85±1.32 | **70.25**±1.22 | 5.2 |
| PerEG (ours) | **72.73**±0.72 | **76.24**±0.58 | **64.32**±0.47 | **61.46**±1.06 | 78.33±3.26 | **77.52**±0.63 | **78.36**±0.68 | 76.25±1.37 | **71.75**±1.17 | **1.9** |

in 5 out of 8 datasets. This demonstrates the effectiveness of our PerEG in learning better graph representation under the unsupervised scenario. In addition, from the ablation study of our PerEG, we can see that the removal of node-oriented discriminator ("w/o $\mathcal{L}_{node}$") and the removal of node-edge-oriented discriminator ("w/o $\mathcal{L}_{edge}$") degrade the performance of our PerEG. This indicates that both node-oriented discriminator and edge-oriented discriminator are important in our PerEG when performing perturbation discrimination. Meanwhile, the removal of factor $\rho$ ("w/o $\rho$") slightly degrades the performance, which implies that our proposed reweighted strategy can improve graph contrastive learning, and lead to better downstream task performance. Further, "w/o $\mathcal{L}_{node}$" performs slightly worse than "w/o $\mathcal{L}_{edge}$", which implies that the node-oriented discriminator can achieve better recognition of node perturbations than the edge-oriented discriminator.

**Semi-supervised Learning** To evaluate the performance of our PerEG in the semi-supervised scenario, we follow the settings in (You et al., 2020; Suresh et al., 2021) and report the experimental results in Table 3. It can be seen that our PerEG performs overall better than the baseline models on all datasets in the semi-supervised learning scenario. This verifies the effectiveness of our PerEG for better graph representation. According to the improved results achieved by comparing with existing models, we can conclude that exploring perturbation discrimination to enhance GCL can make the model perform GCL in a controlled manner, allowing the model to alleviate the introduction of noise augmentation and improve the learning of graph representation.

**Transfer Learning** Following (Hu et al., 2020a; You et al., 2020), we further evaluate the performance of our PerEG in the transfer learning scenario, which pre-trains and finetunes the model in different datasets to evaluate the transferability of the pre-training scheme. The experimental results are reported in Table 4. We can see that our PerEG overall outperforms the baseline models, demonstrating the effectiveness of our PerEG for better graph representation in transfer learning.

### 4.3 WHY CAN PEREG WORK WELL?

**How do the discriminators enhance the representation of augmented graphs?** To analyze how our proposed discriminators enhance the representation of augmented graphs, we show the t-SNE visualization of embeddings of the original graph and the augmentations in Figure 3. The red point denotes the embedding of the original graph, the blue points denote the embeddings of augmentations with discriminators, and the green points denote the embeddings of augmentations without discriminators. It can be seen that The representations of augmented graphs can be normalized around the original graph by means of our proposed discriminators, resulting in better representations for graph contrastive learning.

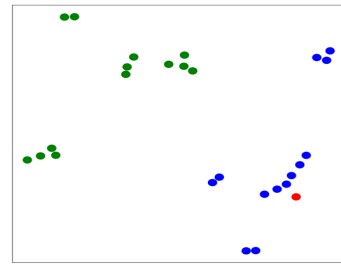

Figure 3: Visualizations of embeddings of the original graph and augmented graphs presented by t-SNE (van der Maaten & Hinton, 2008).

**Visualizations of Graph Embeddings** To qualitatively demonstrate how our PerEG improves the graph representations in down-

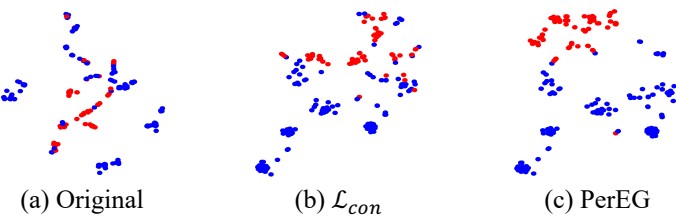

(a) Original       (b) $\mathcal{L}_{con}$       (c) PerEG

Figure 4: Visualizations of the graph embeddings presented by t-SNE (van der Maaten & Hinton, 2008) on MUTAG dataset. "Original" denotes the representations learned by the original embeddings.

stream tasks, Figure 4 shows the t-SNE visualization of graph embeddings learned by the vanilla "original" embeddings (a), the variants of our PerEG "$\mathcal{L}_{con}$" (b), and our PerEG (c). We can observe that the embeddings learned by "Original" largely diffuse and overlap between different labels, which denotes that proposing preferable methods to learn better graph representations is key to improving downstream task performance. In addition, "$\mathcal{L}_{con}$" can produce better graph embeddings in comparison with "Original". This qualitatively demonstrates that contrastive learning in our PerEG can achieve better graph representations. Further, the graph embeddings derived by our PerEG can better separate representations from different labels, which implies that our PerEG can leverage the merits of perturbation discrimination-enhanced contrastive learning, thus obtaining better graph representations.

**How PerEG improves the graph representation?** To illustrate why our PerEG works well in learning graph representations, we conduct experiments by plotting $\mathcal{L}_{align}$ and $\mathcal{L}_{uniform}$ in Figure 5. We train 20 epochs for every model and visualize the checkpoints every 2 epochs. We can see that our PerEG performs significantly better for both alignment and uniformity. This indicates that the perturbation discrimination can improve graph representation in GCL.

## 5 CONCLUSION

In this paper, we propose a novel method to improve graph contrastive learning based on perturbation discrimination, called Perturbation Discrimination-Enhanced graph contrastive learning (PerEG). To be specific, we devise a discriminative model to predict whether each node in the augmented graph was affected by perturbations or not, so as to identify the fine-grained perturbation in a perturbed augmentation compared to the original graph. Then, the discriminative results are

Figure 5: Results of plotting $\mathcal{L}_{align}$ and $\mathcal{L}_{uniform}$ on MUTAG dataset. We train 20 epochs for every model and visualize the checkpoints every 2 epochs. As discussed in (Wang & Isola, 2020), for both $\mathcal{L}_{align}$ and $\mathcal{L}_{uniform}$, models with lower numbers are better.

used to enhance graph contrastive learning by using the perturbed augmentations in a controlled manner. This essentially enables the model to alleviate the introduction of noise augmentation when performing graph contrastive learning, thereby improving the learning of graph representation. Extensive experiments and in-depth analysis in unsupervised, semi-supervised, and transfer learning scenarios verify that our PerEG achieves state-of-the-art performance in learning generalizability and robustness graph representations on eight diverse graph-structured datasets.

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

## A    NOTATIONS USED IN THIS PAPER

The notations used in this work are listed in Table 5.

Table 5: Notations used in this paper.

| Notation | Corresponding Meanings |
|---|---|
| $\mathcal{G}$ | The original graph |
| $\mathcal{G}'$ | An augmentation of the original graph $\mathcal{G}$ |
| $\mathcal{T}(\mathcal{G})$ | An augmentation pool for graph $\mathcal{G}$ |
| $f_g(\cdot;\theta)$ | A GNN encoder with parameters $\theta$ |
| $g(\cdot)$ | The Projection Head |
| $\mathcal{V}$ | The set of nodes of graph $\mathcal{G}$ |
| $\mathcal{E}$ | The set of edges of graph $\mathcal{G}$ |
| $\boldsymbol{X}_v$ | The matrix of node features |
| $\boldsymbol{X}_e$ | The matrix of edge features |
| $d_v$ | The dimensionality of node feature |
| $d_e$ | The dimensionality of edge feature |
| $h_v$ | The representation of a node $v$ |
| $h_{\mathcal{G}}$ | The representation of a graph $\mathcal{G}$ |
| $h_v^{(k)}$ | The representation $h_v$ in the $k$-th layer GNN |
| AGGREGATION$(\cdot)$ | A trainable function that aggregates messages from the neighbors of node $v$ |
| $\mathcal{N}(v)$ | The set of neighbors of node $v$ |
| UPDATE$(\cdot)$ | A trainable function that updates the representation of node $v$ |
| POOL$(\cdot)$ | The pooling function that generates the representation $h_{\mathcal{G}}$ |
| $\mathcal{D}_n$ | The Node-oriented Discriminator |
| $\mathcal{D}_e$ | The Edge-oriented Discriminator |
| $\mathcal{L}_{node}$ | The predictive loss of the Node-oriented Discriminator |
| $\mathcal{L}_{edge}$ | The predictive loss of the Edge-oriented Discriminator |
| $\mathcal{P}_n(v)$ | An indicator to determine whether node $v$ is disturbed |
| $\mathcal{P}_e(v)$ | An indicator to determine whether node $v$ is affected by the edges' perturbations |
| $z_i$ | The representation of graph $\mathcal{G}_i$ after Projection Head $g(\cdot)$ |
| $z_i'$ | The positive augmentation representation for $z_i$ |
| $\mathcal{B}$ | The minibatch |
| sim$(\cdot)$ | The cosine similarity function |
| $\ell(\mathcal{G}_i)$ | The NT-Xent for graph $\mathcal{G}_i$ |
| $\mathcal{L}_{con}$ | The contrastive loss |
| $\lambda_1, \lambda_2, \lambda_3$ | Scaling weights to balance the losses |
| $\rho_n, \rho_e$ | The reweighted factor |
| $\gamma_n, \gamma_e$ | Scaling weights to balance $\rho_n$ and $\rho_e$ |

## B    ALGORITHM OF OUR PerEG

For a dataset containing $M$ graphs: Data $\{\mathcal{G}_m : m \in M\}$, we use our Perturbation Discrimination-Enhanced Graph Contrastive Learning (PerEG) to train the GNN encoder $f_g(\cdot;\theta)$ for producing semantically good graph representation. The procedure of our PerEG is depicted in Algorithm 1.

## C    BASELINES

We compare and evaluate our PerEG framework with a series of baseline models, which are grouped by graph kernel methods, graph self-supervised learning methods, predictive learning methods, and contrastive learning methods. The detailed introduction of the baseline models is as follows:

1) graph kernel methods:

- **GL** (Shervashidze et al., 2009). A graph kernel method that measures the similarity between graphs based on counting small subgraphs. It can be used for graph classification and clustering tasks. The kernel computes a similarity score based on the frequency of graph occurrences, providing a way to compare the structural characteristics of different graphs.

---

**Algorithm 1:** Perturbation Discrimination-Enhanced Graph Contrastive Learning

---
**Initialize:** Data $\{\mathcal{G}_m : m \in M\}$, GNN encoder $f_g(\cdot; \theta)$, pooling function POOL$(\cdot)$, projection head $g(\cdot)$, node-oriented predictor $\mathcal{D}_n(\cdot; \theta_n)$, edge-oriented predictor $\mathcal{D}_e(\cdot; \theta_e)$, structure-perturbed augmentation pool $\mathcal{T}$

**1 for** sampled minibatch $\mathcal{B}$ of data $\{\mathcal{G}_i : i \in N_b\}$ **do**

**2**     **for** $\mathcal{G}_i = \{\mathcal{V}, \mathcal{E}\} \in \mathcal{B}$ **do**

**3**        Sample $\mathcal{G}'_i$ form $\mathcal{T}$: $\mathcal{G}'_i \leftarrow \mathcal{T}(\mathcal{G}_i)$ # Structure-perturbed augmentation

**4**        $\{h_v\}_{v=1}^{|\mathcal{V}|} = f_g(\mathcal{G}_i; \theta)$

**5**        $\{h'_v\}_{v=1}^{|\mathcal{V}|} = f_g(\mathcal{G}'_i; \theta)$

**6**        $z_i = g\Big(\text{POOL}\big(\{h_v\}_{v=1}^{|\mathcal{V}|}\big)\Big)$

**7**        $z'_i = g\Big(\text{POOL}\big(\{h'_v\}_{v=1}^{|\mathcal{V}|}\big)\Big)$

**8**     **end**

**9**     $\mathcal{L}_{node}(\mathcal{G}_i) = \sum_{v=1}^{|\mathcal{V}|}\Big(-\mathbb{1}\big(\mathcal{P}_n(v)=1\big)\log\big(\mathcal{D}_n(h'_v; \theta_n)\big) - \mathbb{1}\big(\mathcal{P}_n(v)=0\big)\log\big(1 - \mathcal{D}_n(h'_v; \theta_n)\big)\Big)$

**10**     $\mathcal{L}_{edge}(\mathcal{G}_i) = \sum_{v=1}^{|\mathcal{V}|}\Big(-\mathbb{1}\big(\mathcal{P}_e(v)=1\big)\log\big(\mathcal{D}_e(h'_v; \theta_e)\big) - \mathbb{1}\big(\mathcal{P}_e(v)=0\big)\log\big(1 - \mathcal{D}_e(h'_v; \theta_e)\big)\Big)$

**11**     $\rho_n = \frac{1}{|\mathcal{V}_{pn}|}\sum_{v \in \mathcal{V}_{pn}}\Big(\mathbb{1}(\mathcal{D}_n(v)=1)\Big)$

**12**     $\rho_e = \frac{1}{|\mathcal{V}_{pe}|}\sum_{v \in \mathcal{V}_{pe}}\Big(\mathbb{1}(\mathcal{D}_e(v)=1)\Big)$

**13**     $\rho = \gamma_n\rho_n + \gamma_e\rho_e$

**14**     $\ell(\mathcal{G}_i) = -\log\frac{\exp(\text{sim}(z_i, z'_i)/\tau)}{\sum_{k=1, k \neq i}^{N_b}\exp(\text{sim}(z_i, z_k)/\tau)} \times \rho$

**15**     $\mathcal{L}_{node} = \frac{1}{N_b}\sum_{\mathcal{G}_i \in \mathcal{B}}\mathcal{L}_{node}(\mathcal{G}_i)$ # Loss of Node-oriented Predictor

**16**     $\mathcal{L}_{edge} = \frac{1}{N_b}\sum_{\mathcal{G}_i \in \mathcal{B}}\mathcal{L}_{edge}(\mathcal{G}_i)$ # Loss of Edge-oriented Predictor

**17**     $\mathcal{L}_{con} = \frac{1}{N_b}\sum_{\mathcal{G}_i \in \mathcal{B}}\ell(\mathcal{G}_i)$ # Loss of Graph-level Contrastive Learning

**18**     $\mathcal{L} = \lambda_1\mathcal{L}_{con} + \lambda_2\mathcal{L}_{node} + \lambda_3\mathcal{L}_{edge}$

**19**     Update $f_g(\cdot; \theta)$, $g(\cdot)$, $\mathcal{D}_n(\cdot; \theta_n)$, and $\mathcal{D}_e(\cdot; \theta_e)$ to minimize $\mathcal{L}$

**20 end**

**21 Return:** GNN Encoder $f_g(\cdot; \theta)$

---

- **WL** (Shervashidze et al., 2011). A Weisfeiler-Lehman sub-tree kernel measures the similarity between graphs by comparing their labeled substructures. It iteratively refines the node labels by aggregating the labels of neighboring nodes, capturing graph structure, and preserving information about node attributes.

- **DGK** (Yanardag & Vishwanathan, 2015). A deep graph kernel method that uses neural networks to learn representations of subgraphs and combine them to compute a similarity score, enabling more expressive and flexible graph comparisons for various graph analysis tasks.

2) graph self-supervised learning methods:

- **GAE** (Kipf & Welling, 2016). A graph neural network model used for unsupervised learning on graph-structured data. It aims to learn low-dimensional latent representations of nodes in a graph by reconstructing the adjacency matrix or node attributes, enabling tasks such as link prediction and node classification in the graph domain.

- **node2vec** (Grover & Leskovec, 2016). An algorithm for learning feature representations or embeddings for nodes in a graph. It combines random walks and the Skip-gram model to capture structural properties and node relationships. The resulting embeddings enable various graph analysis tasks such as link prediction, node classification, and clustering.

- **sub2vec** (Adhikari et al., 2018). An unsupervised algorithm that has a global view of the learning subgraph and captures the similarities and differences between the properties of the entire subgraph.

- **graph2vec** (Narayanan et al., 2017). A graph representation learning algorithm that uses the Skip-gram model from Word2Vec to learn continuous embeddings for entire graphs.

By aggregating node embeddings, Graph2Vec captures the structural properties and global characteristics of graphs, enabling downstream graph analysis tasks.

- **InfoGraph** (Sun et al., 2020). Obtaining graph representation by maximizing the mutual information between the graph-level representation and the representation of substructures at different scales (such as nodes, edges, and triangles).

3) predictive learning methods:

- **ContextPred** (Hu et al., 2020a). Performing node-level self-supervised pre-training and graph-level multi-task supervised pre-training. When GNN pretraining is completed, the pre-trained GNN model is fine-tuned for downstream tasks, specifically adding a linear classifier to predict downstream graph labels based on graph-level representation.

4) contrastive learning methods:

- **MVGRL** (Hassani & Khasahmadi, 2020). Performing contrastive learning on different structural views of graphs for both node and graph levels to enrich feature learning.

- **GraphCL** (You et al., 2020). Designing four types of graph augmentations to learn the similarity metrics between augmentations with contrastive learning.

- **JOAO** and **JOAOv2** (You et al., 2021). Utilizing an auto augmentation strategy to select the best augmentations from the four types of graph augmentations designed by GraphCL.

- **AD-GCL-FIX** and **AD-GCL-OPT** (Suresh et al., 2021). Learning graph augmentation strategies with adversarial methods.

- **SimGRACE** (Xia et al., 2022a). Generating contrastive objects of graphs for contrast by perturbing the weights of the model.

- **AutoGCL** (Yin et al., 2022). Designing a view generation learner for each graph augmentation method to learn the probability distribution of the augmentation method for specific graph data.

- **GPA** (Zhang et al., 2024). Let each graph to select its own suitable data augmentation operations through a learnable augmentation selector.

## D  MORE DETAILS OF EXPERIMENTAL IMPLEMENTATION

All experiments are conducted in Intel(R) Xeon(R) Gold 5220R CPU @ 2.20GHz, 4 NVIDIA V100 GPU with 32 GB of RAM. The detailed settings of hyperparameters in our unsupervised and semi-supervised learning experiments are shown in Table 6.

## E  ANALYSIS OF AUGMENTATION STRENGTH

To examine whether the strength of augmentations can affect the performance of our PerEG, we conduct experiments with different strengths in producing augmentations and show the results in Figure 6. The structure-perturbed graph augmentation results of Figure 6(a)-(c) show that ratios near 0.5 worsen the performance, denoting that disturbing around half of the nodes in a graph will increase the uncertainty of the graph structure, thereby increasing the difficulty of prediction. Figure 6(d) shows that, for edge perturbation, the fluctuation in performance is relatively slight at different ratios, implying that an edge perturbation affects multiple nodes, and a smaller proportion of edge perturbations may cause significant structural changes.

Further, Figure 7 shows the comparison results of using different perturbation strategies to produce perturbed augmentations. Our PerEG achieves better performance than GraphCL in different augmentation strategies. Meanwhile, the improvement of different strategies varies. Therefore, we need to explore different augmentation strategies to learn richer graph representations.

Table 6: Experimental implementation for unsupervised and semi-supervised learning.

| Hyperparameter | Description | Unsupervised | Semi-supervised | |
| --- | --- | --- | --- | --- |
| | | | Pre-train | Fine-tune |
| - | Learning rate | 0.005 | 0.001 | 0.001 |
| $\mathcal{B}$ | Batch size | 128 | 128 | 128 |
| $K$ | Number of GNN layers | 3 | - | - |
| $d_h$ | Hidden layer dimensionality | 32 | 128 | 128 |
| $\lambda_1$ | Scaling weight of contrastive loss $\mathcal{L}_{con}$ | 1.0 | 1.0 | - |
| $\lambda_2$ | Scaling weight of node predictive loss $\mathcal{L}_{node}$ | 1.0 | 1.0 | - |
| $\lambda_3$ | Scaling weight of edge predictive loss $\mathcal{L}_{edge}$ | 0.5 | 0.5 | - |
| $\gamma_n$ | Scaling weight of node reweighted factor $\rho_n$ | 0.5 | 0.5 | - |
| $\gamma_e$ | Scaling weight of edge reweighted factor $\rho_e$ | 0.5 | 0.5 | - |
| - | Training epochs | 20 | 25 | 60 |
| $\eta_a$ | Augmentation ratio | 0.2 | 0.2 | - |
| $POOL(\cdot)$ | Pooling function | 'sum' | 'sum' | 'sum' |
| - | Number of feature transform layers | - | 1 | 1 |
| - | Number of GCN layers | - | 3 | 3 |
| - | Number of fully-connect layers | - | 2 | 2 |
| - | Learning rate of fine-tuning | - | - | 0.001 |
| - | Training epochs of fine-tuning | - | - | 20 |
| - | Number of k-fold splits in fine-tuning | - | - | 10 |

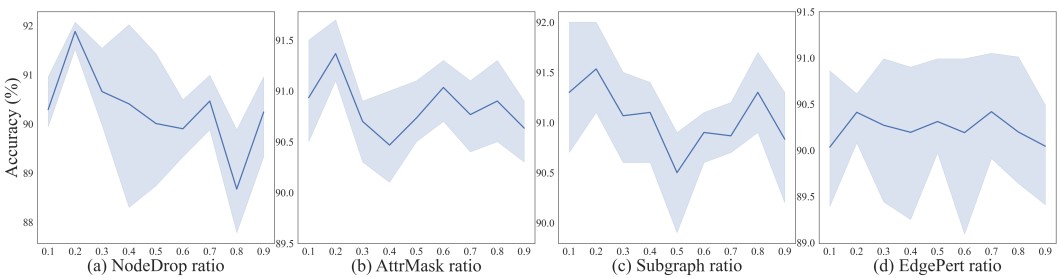

Figure 6: Experimental results of performance versus augmentation strength on MUTAG dataset.

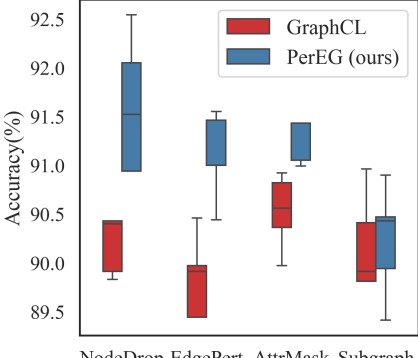

Figure 7: Comparison results in different perturbation strategies.

# F ROBUSTNESS STUDY

Following Guerranti et al. (2023), we conducted additional experiments of robustness studies for our model. The results are shown in Table 7. From the results, we can see that our model (PerEG) can succeed in enhancing adversarial robustness in comparison with GraphCL.

Table 7: Robustness studies for PerEG on TUDataset.

| Methods | PROTEINS | NCI1 | DD |
|---|---|---|---|
| GraphCL (Clean) | 65.93 | 73.09 | 68.85 |
| GraphCL (Attack) | 40.29 ($\downarrow$ 38.89) | 32.20 ($\downarrow$ 55.94) | 15.74 ($\downarrow$ 75.34) |
| PerEG (Clean) | 68.73 | 74.03 | 74.83 |
| PerEG (Attack) | 48.07 ($\downarrow$ 32.69) | 33.35 ($\downarrow$ 54.95) | 26.20 ($\downarrow$ 64.99) |

## G   FURTHER STUDIES ON THE PERFORMANCE OF DISCRIMINATORS

In order to further validate the performance of the discriminator we used, we analyze the accuracy of the two discriminators in PerEG. Here we report the performances of the discriminators in Table 8. We can see that the discriminators perform well on all datasets, the accuracy of all datasets is above 80%. This proves that the discriminators are effective and capable of identifying most of the perturbations. Further, those small parts that cannot be recognized, will be considered as noise, making them ineffective in graph contrastive learning.

Table 8: Accuracy of the discriminators (%)

| Discriminator | NCI1 | PROTEINS | DD | MUTAG | COLLAB | RDT-B | RDT-M5K | IMDB-B |
|---|---|---|---|---|---|---|---|---|
| $\mathcal{D}_n$ | 87.23 | 85.45 | 90.32 | 90.89 | 86.43 | 90.20 | 81.83 | 88.26 |
| $\mathcal{D}_e$ | 87.87 | 86.02 | 92.75 | 91.43 | 87.27 | 91.56 | 85.55 | 89.34 |

Additionally, we combine the two discriminators and then retest the performance of the model. As shown in Table 9, the performance of combining two discriminators is worse than our PerEG.

Table 9: Comparison results of combining two discriminators and our PerEG (%)

| Methods | NCI1 | PROTEINS | DD | MUTAG | COLLAB | RDT-B | RDT-M5K | IMDB-B |
|---|---|---|---|---|---|---|---|---|
| Combine | 78.64 | 74.86 | 79.15 | 89.35 | 73.02 | 89.85 | 56.30 | 71.67 |
| PerEG | 79.94 | 76.04 | 80.02 | 90.61 | 73.84 | 90.57 | 56.81 | 72.97 |

## H   EXPERIMENTS ON NODE CLASSIFICATION TASK

We also evaluate our PerEG in the unsupervised representation learning in the node classification scenario following Veličković et al. (2019) and You et al. (2020). We use GCN as the GNN-based encoder to generate the node embeddings and then feed them into a downstream classifier. The results in Table 10 show that PerEG outperforms both DGI and GraphCL, proving the superiority of our PerEG.

## BROADER IMPACTS

Learning on graph-structured data has a wide range of interests and applications, such as recommendation systems, social media, neural structure search, and drug discovery. Our PerEG contributes a general framework for processing different kinds of graph augmentations. Further, the proposed node-level predictive learning can perceive the fine-grained structural differences and identify the topological information of the graph from node-oriented and edge-oriented perspectives, providing a new idea for learning graph-structured information.

## LIMITATIONS

In order to mitigate the generation of isolated outlier nodes, we produce four perturbed augmentations for each graph. However, there is still a situation where a node is isolated in different augmentations, which will hinder the performance of perturbation discrimination in our PerEG. Therefore, in future work, we will consider trying more reasonable strategies for producing perturbed augmentations to avoid the generation of isolated nodes, aiming to further improve the performance of our PerEG.

Table 10: Comparing classification accuracy on top of learned node representations. The compared DGI(Veličković et al., 2019) and GraphCL(You et al., 2020) performance are from the original paper under the same experiment setting.

| Methods | Augmentation | Cora | Citeseer |
|---|---|---|---|
| DGI | - | 82.30±0.60 | 71.80±0.70 |
| GraphCL | NodeDrop v.s. NodeDrop | 81.76±0.17 | 73.14±0.15 |
| | EdgePert v.s. EdgePert | 82.32±0.15 | 73.11±0.19 |
| | AttrMask v.s. AttrMask | 81.78±0.17 | 72.05±0.22 |
| | Subgraph v.s. Subgraph | 81.71±0.14 | 73.12±0.17 |
| | NodeDrop v.s. Identical | 82.41±0.10 | 72.22±0.18 |
| | EdgePert v.s. Identical | 82.45±0.11 | 72.23±0.17 |
| | AttrMask v.s. Identical | 82.45±0.12 | 72.31±0.13 |
| | Subgraph v.s. Identical | 82.49±0.12 | 72.33±0.18 |
| PerEG (ours) | NodeDrop v.s. NodeDrop | 81.90±0.16 | 73.70±0.20 |
| | EdgePert v.s. EdgePert | 83.58±0.15 | 74.11±0.16 |
| | AttrMask v.s. AttrMask | 82.78±0.21 | 72.69±0.25 |
| | Subgraph v.s. Subgraph | 82.26±0.12 | 73.69±0.20 |
| | NodeDrop v.s. Identical | 83.16±0.13 | 73.53±0.26 |
| | EdgePert v.s. Identical | 83.04±0.18 | 72.76±0.20 |
| | AttrMask v.s. Identical | 83.04±0.19 | 72.72±0.27 |
| | Subgraph v.s. Identical | 83.06±0.13 | 72.80±0.20 |

