# OpenReview forum: "Enhance Graph Contrastive Learning with Perturbation Discrimination"
_ICLR.cc/2025/Conference — ICLR 2025 Conference Withdrawn Submission_

### Official Review · Reviewer_tKDS · 2024-10-29

**Soundness:** 2
**Presentation:** 2
**Contribution:** 2
**Rating:** 3
**Confidence:** 5

**Summary:**

Self-supervised learning on graph-structured data aims to create representations that are effective for downstream tasks. Graph contrastive learning (GCL), often based on data augmentation, has shown promise in this area. However, certain augmentations can disrupt graph semantics, causing GCL methods to suffer from noise. To address this, the authors propose a model called Perturbation Discrimination-Enhanced GCL (PerEG), which trains a discriminative model to identify perturbed nodes in augmented graphs. This helps the GCL framework selectively use augmentations, minimizing noise and enhancing performance. Experiments show that PerEG outperforms current methods across multiple datasets in unsupervised, semi-supervised, and transfer learning contexts.

**Strengths:**

1. This paper gives a clear and detailed description of the algorithm.
2. This paper addresses the problems of Graph Contrastive Leaning on both graph classification and node classification tasks.
3. This paper provides extensive experiments that validate the performance of PerEG.

**Weaknesses:**

1. The proposed method introduces discriminative components to Graph Contrastive Learning (GraphCL), which seems to represent a contribution of the components rather than the framework itself. Therefore, it is necessary to apply these components to other Graph Contrastive Learning algorithms. However, the evaluation has only been conducted on GraphCL. I recommend applying the Perturbation Discrimination components to other algorithms, such as AD-GCL, AutoGCL, and GPA, to assess their effectiveness.


2. The baselines differ between the settings of Unsupervised, Semi-supervised, and Transfer Learning, but the authors do not explain why certain algorithms present in the Unsupervised setting are absent in the Semi-supervised and Transfer Learning settings. Please provide an explanation for these differences.


3. The paper lacks hyper-parameter sensitivity experiments, particularly for parameters such as $\lambda_1, \lambda_2, \lambda_3$, Even though the authors mention “optimal hyper-parameters in the pilot studies”, there are no details provided about these pilot studies. Please include the details about pilot studies to enhance the understanding of the hyper-parameter selection process.


4. The authors provide components to control the augmentations; however, many graph contrastive learning models also focus on controlling augmentations, such as spectral-based methods [4, 5, 6], community-based methods [1, 4], and saliency-related methods [7]. These should be discussed in the literature review for a more comprehensive understanding of the field.


5. The section on related work regarding Graph Contrastive Learning (GCL) is insufficient, as it primarily discusses methods developed before 2023. Additionally, there is no comparison between your work and other recent methods. Please include a broader range of recent graph contrastive learning works [1-7] and provide detailed comparisons between your approach and these related works.


6. The source code is lacking.

[1] Clusterscl: Cluster-aware supervised contrastive learning on graphs, WWW, 2022.

[2] Graph Self-supervised Learning with Augmentation-aware Contrastive Learning. WWW 2023.

[3] Simple and Asymmetric Graph Contrastive Learning without Augmentations. NeurIPS 2023

[4] Community-Invariant Graph Contrastive Learning, ICML, 2024.

[5] Spectral Feature Augmentation for Graph Contrastive Learning and Beyond, AAAI, 2023.

[6] Spectral augmentation for self-supervised learning on graphs, ICLR, 2023.

[7] Boosting graph contrastive learning via graph contrastive saliency, ICML, 2023.

**Questions:**

1. In the discriminative components, how do you determine whether the semantics have changed after augmentation? Your proposed components only identify whether nodes or edges are influenced by the augmentation, but being influenced does not necessarily imply that their semantics have changed. Could you elaborate on how you determined whether the semantics changed or not?


2. In Equation 3, if nodes are dropped, there would be no representations available. However, in the paragraph titled "Node-oriented Discriminator", it is stated that the loss can still be calculated even when nodes are dropped. How do you address such situations? Please provide more details on how this is managed.

3. The rationale for selecting these baselines in Sec 4.1 is not discussed. Please provide a justification for why these baselines were chosen.


4. The comparison in the node classification task is not entirely fair, as GraphCL is not specifically designed for node classification. I recommend including some Graph Contrastive Learning (GCL) baselines that are focused on node classification to provide a more accurate evaluation.


5. The proposed method introduces discriminative components to Graph Contrastive Learning (GraphCL), which seems to represent a contribution of the components rather than the framework itself. Therefore, it is necessary to apply these components to other Graph Contrastive Learning algorithms. However, the evaluation has only been conducted on GraphCL. I recommend applying the Perturbation Discrimination components to other algorithms, such as AD-GCL, AutoGCL, and GPA, to assess their effectiveness.

6. The paper lacks details regarding the robustness study. Please provide more information on this aspect. Additionally, consider including more baselines beyond just GraphCL to enhance the evaluation of your method.

Overall, it is a fair paper, but not well organized, and not well-polished. But if the author fixes my concerns in Weaknesses and Questions, I could raise the rating.

---

### Official Review · Reviewer_eaDk · 2024-10-30

**Soundness:** 2
**Presentation:** 3
**Contribution:** 2
**Rating:** 5
**Confidence:** 4

**Summary:**

This paper introduces Perturbation Discrimination-Enhanced Graph Contrastive Learning (PerEG), a novel approach aimed at enhancing graph contrastive learning by controlling noise in data augmentations. By using a discriminative model with node-oriented and edge-oriented discriminators, PerEG predicts and refines perturbations at the node and edge levels, ensuring that augmentations retain semantic alignment with the original graph. Through extensive experiments, PerEG demonstrates superior performance in unsupervised, semi-supervised, and transfer learning tasks across various datasets, outperforming state-of-the-art methods in graph representation learning.

**Strengths:**

- The paper is well-written, clear, and easy to follow.
- The idea of discriminating graph perturbations is innovative and well-motivated.
- The experiments demonstrate the effectiveness of the proposed model, and visualizations provide an intuitive understanding of its impact.

**Weaknesses:**

- The paper lacks evidence on to what extent do perturbations enhance GCL performance and to what extend do perturbations hurt GCL performance. Providing insights into this quantitatively would improve the paper's clarity and practical value.
- The model involves numerous hyperparameters to tune, including key parameters $\gamma_n$, $\gamma_e$, $\lambda_1$, $\lambda_2$, and $\lambda_3$, which could limit its generalizability and increase tuning complexity.
- An in-depth analysis of the discriminators' performance would be helpful. Specifically, how effective are they at discriminating perturbed nodes/edges, and how does this capability relate to overall GCL performance?
- Constructing the ground truth to train the node-oriented and edge-oriented discriminators may be computationally costly. Each graph is augmented in each training epoch, requiring ground truth calculations for the discriminators in every instance. The authors are encouraged to provide a complexity analysis and plot the training time overhead to clarify this aspect.
- The motivation behind certain design choices in the model is not entirely clear. For example, in Equation 5, it is unclear why the model would benefit from a higher proportion of perturbed nodes or edges. The authors should offer more insights into this decision.
- The performance improvements reported are relatively modest. Expanding the experimental evaluation to additional datasets would provide a clearer picture of the model's robustness and general applicability.

**Questions:**

Please refer to weaknesses above.

---

### Official Review · Reviewer_mxUb · 2024-11-03

**Soundness:** 2
**Presentation:** 2
**Contribution:** 2
**Rating:** 5
**Confidence:** 5

**Summary:**

The paper proposes a GCL framework with two discriminators for node and edge to discriminate whether the graph is overly perturbed or not.

**Strengths:**

1. The paper clearly presents the proposed method. Overall, the paper is well organized.

2. Many experiments are conducted to demonstrate the effectiveness of the proposed method.

**Weaknesses:**

1. At line 228, the authors mention that the discriminator is trained to predict whether nodes are dropped or perturbed. However, if a node is dropped, there would be no corresponding node representation. In this case, why is it necessary for the discriminator to make this prediction, and how can it accomplish this task?

2. Although the authors introduce two discriminators to assess whether the graph augmentation is excessively aggressive, the proposed design does not inherently guides the strength of the graph augmentation. This raises the possibility that the discriminator could make correct predictions for all nodes and edges, causing the framework to degrade into a standard Graph Contrastive Learning (GCL) method. The authors should analyze the likelihood of such degradation and clarify the implications of their method if this degradation occurs.

3. The authors are encouraged to conduct experiments demonstrating the correlation between the discrimination accuracy and the final performance outcomes.

**Questions:**

Please refer to the weakness section.

---

### Note · Authors · 2025-01-23

I have read and agree with the venue's withdrawal policy on behalf of myself and my co-authors.